# The Use of Mobile Health Interventions for Outcomes among Middle-Aged and Elderly Patients with Prediabetes: A Systematic Review

**DOI:** 10.3390/ijerph192013638

**Published:** 2022-10-20

**Authors:** Yaltafit Abror Jeem, Russy Novita Andriani, Refa Nabila, Dwi Ditha Emelia, Lutfan Lazuardi, Hari Koesnanto

**Affiliations:** 1Faculty of Medicine, Public Health and Nursing, Universitas Gadjah Mada, Yogyakarta 55281, Indonesia; 2Faculty of Medicine, Universitas Islam Indonesia, Yogyakarta 55584, Indonesia; 3Department of Health Policy and Management, Faculty of Medicine, Public Health and Nursing, Universitas Gadjah Mada, Yogyakarta 55281, Indonesia; 4Department of Family and Community Medicine, Faculty of Medicine, Public Health and Nursing, Universitas Gadjah Mada, Yogyakarta 55281, Indonesia

**Keywords:** prediabetic state, telemedicine, m-Health, middle-aged, elderly, intervention, systematic review

## Abstract

Background: There are currently limited systematic reviews of mobile health interventions for middle-aged and elderly patients with prediabetes from trial studies. This review aimed to gather and analyze information from experimental studies investigating the efficacy of mobile health usability for outcomes among middle-aged and elderly patients with prediabetes. Methods: We conducted a literature search in five databases: Clinicaltrials.gov, the International Clinical Trials Registry Platform (ICTRP), PubMed, ProQuest, and EBSCO, with a date range of January 2007 to July 2022 written in English, following a registered protocol on PROSPERO (CRD42022354351). The quality and possibility of bias were assessed using the Jadad score. The data extraction and analysis were conducted in a methodical manner. Results: A total of 25 studies were included in the qualitative synthesis, with 19 studies using randomized trial designs and 6 studies with non-randomized designs. The study outcomes were the incidence of diabetes mellitus, anthropometric measures, laboratory examinations, measures of physical activity, and dietary behavior. During long-term follow-up, there was no significant difference between mobile health interventions and controls in reducing the incidence of type 2 diabetes. The findings of the studies for weight change, ≥3% and ≥5% weight loss, body mass index, and waist circumference changes were inconsistent. The efficacy of mobile health as an intervention for physical activity and dietary changes was lacking in conclusion. Most studies found that mobile health lacks sufficient evidence to change hbA1c. According to most of these studies, there was no significant difference in blood lipid level reduction. Conclusions: The use of mobile health was not sufficiently proven to be effective for middle-aged and elderly patients with prediabetes.

## 1. Introduction

Impaired Fasting Glucose (IFG), Impaired Glucose Tolerance (IGT), or both, and/or elevated levels of Hemoglobin A1c (HbA1c) are all signs of prediabetes [1]. In 2015, there were over 318 million cases of IGT worldwide, and by 2040, there are expected to be about 481 million cases [2]. Prediabetes has become more common as people become older [3]. The transition from normoglycemia to prediabetes or diabetes, or from prediabetes to diabetes, was prevalent in middle-aged and elderly people, suggesting the need for continuing treatment [4]. Individuals with prediabetes are four times more likely than individuals with normal glucose tolerance to develop Type-2 Diabetes Mellitus (T2DM) [5]. Prediabetes can lead to major complications, whether or not patients have T2DM. Microalbuminuria is about twice as common in patients with prediabetes as it is in healthy people, and it can lead to significant problems such as chronic kidney disease (CKD) and macrovascular disease. Prediabetes, on the other hand, can cause many types of diabetic neuropathy, including peripheral neuropathy, polyneuropathy, small-fiber neuropathy, and autonomic neuropathy [6]. 

Changes in lifestyle, such as daily physical activity and a healthy diet, can reduce the risk of prediabetes, improve the health of diabetic patients, and prevent complications [7]. Consultations with health care professionals at medical facilities are commonly used to modify a person’s lifestyle. Personal coaching for physical activity and personal consultations for healthy eating recommendations are also not generally offered. As a result, mobile health or m-Health interventions could be some of the solutions for improving self-management awareness and compliance with prediabetes interventions including lifestyle changes and healthy foods [8]. 

Mobile phone ownership is on the rise, with an estimated 90% of the world’s population having one by 2020. According to the most recent statistics, 80% of older adults own a mobile phone (aged 65 years and older) [9]. As the number of mobile phone owners increases, so does the number of mobile phone software applications (apps), including mobile health apps [10]. Health-related mobile phone apps are more preferable since they are less expensive, handier, and more engaging [11]. Their use has been applied in both developed and developing countries, and results demonstrated that they can provide great promise when it comes to providing individualized medical recommendations [8].

Previous systematic reviews on the efficacy of mobile health interventions to improve health outcomes concluded that interventions have the potential to improve health outcomes as well as diet, physical activity, and clinical metabolic parameters [12,13,14]. There were various types of mobile health interventions to achieve positive health outcome change [15,16], such as app-based mobile interventions [16,17,18,19], text message-based interventions [16,20,21,22], website-based interventions [23,24], wearable devices [16,25,26], and so on. Several studies [15,18,23] suggested that future research be conducted to determine the best number and combination of different technology features to maximize intervention efficacy, utilization patterns, and participant engagement. Despite this, evidence on the effectiveness of mobile health interventions for some chronic conditions and non-communicable diseases has been inconsistent and remains weak [27,28].

There are currently limited systematic reviews of mobile health interventions from trial studies for middle-aged and elderly patients with prediabetes [29,30,31]. Elavsky et al. [31] published a systematic review of interventions utilizing mobile technology to change physical activity, sedentary behavior, and sleep among adults aged 50 and older. The review did not include a specific focus on prediabetes management through mobile health interventions and health outcomes. Inconsistent findings, effects on health outcomes, and intervention details remain to be investigated [29,31]. Therefore, the goal of this study was to compile and analyze evidence from experimental studies investigating the efficacy of mobile health usability for outcomes among middle-aged and elderly patients with prediabetes.

## 2. Methods

### 2.1. Study Design and Research Sample

This systematic review was conducted and reported in accordance with the Preferred Reporting Items for Systematic Reviews and Meta-Analyses (PRISMA) 2020 statement (Appendix A) [32]. The protocol has been registered in the Prospective Registry of Systematic Reviews (PROSPERO) database. The registration number is CRD4202235435. We used PubMed, ProQuest, EBSCO, Clinicaltrials.gov, and the International Clinical Trials Registry Platform (ICTRP) to conduct a systematic search. The scope of the search was limited to English-language publications published between January 2007 and July 2022. Duplicate records were eliminated, and all records were imported into Mendeley Desktop from the database search. 

The following keywords were used to search the database: ((prediabetes AND (minimize risk factor OR physical activity OR body mass index OR weight loss OR healthy diet OR smoking OR alcohol consumption) AND (mobile phone application OR mobile phone app OR telemedicine OR telehealth OR mobile health OR health, mobile OR health OR mobile) AND (middle-aged OR elderly)). 

### 2.2. Inclusion and Exclusion Criteria

The following were the inclusion criteria for the studies that were included: (1) Participants had prediabetes according to American Diabetes Association (ADA) criteria, which included impaired fasting glucose (IFG) 5.6–6.9 mmol/L, impaired glucose tolerance (IGT) 7.8–11.0 mmol/L, or HbA1c (hemoglobin A1c) 5.7–6.4% without a history of T2DM and diabetes medications (insulin or oral anti hyperglycemic agents); (2) Participants were 40 years of age or older; (3) Mobile health intervention is a major component of the study; (4) the mobile health provided lifestyle modification (e.g., physical activity and healthy diets); (5) the study measured participants’ glycated hemoglobin (HbA1c), weight, fasting blood sugar, body mass index(BMI), or other health-related outcomes; (6) the study design is a randomized controlled trial (RCT) or other types experimental study; and (7) the full-text of the article was available. We excluded studies that: (1) Participants had gestational diabetes mellitus (GDM); (2) the study was not a primary study; (3) protocol publication; (4) and did not meet one or more of the inclusion criteria.

### 2.3. Study Selection

Three reviewers (Y.A.J., R.N. and D.D.E.) worked independently to identify articles by looking at the titles, then studying the abstracts, and finally reading the full-text form. The full articles of any possibly relevant studies were then obtained for a thorough review by all reviewers. In the event of a disagreement, a consensus decision was achieved.

### 2.4. Data Extraction

Three reviewers (Y.A.J., R.N. and D.D.E.) retrieved data from the included studies and entered it into a spreadsheet independently. Author, year, country, sample size, study design, details of intervention and control, outcomes of interest, and the other key results were among the data retrieved from the studies. If the needed data were not reported in an article, we contacted the article’s corresponding author to obtain any missing information. If within four weeks of being contacted, the corresponding author does not respond, then the study was not included.

### 2.5. Quality Assessment

Using an adapted version of the Jadad score, two reviewers (R.N. and D.D.E.) independently assessed the quality and risk of bias of the included studies [33]. In the event of a disagreement, another author (Y.A.J.) joined the debate to assist in the resolution of the issues.

### 2.6. Data Synthesis

The Cochrane methodologies for data analysis and syntheses [34] were used to accomplish the data synthesis (narrative synthesis). Data from both qualitative and quantitative sources were described, analyzed, and classified individually. The characteristics of the research, as well as the significant disparities between them, were meticulously recorded. Topics that were relevant were highlighted, and the data were translated into a descriptive style. As stated in the original studies, the outcome results are described in detail. Due to a paucity of studies with similar settings, interventions and results, a meta-analysis could not be performed.

## 3. Results

### 3.1. Search Results

For our investigation, we selected five electronic search engines (PubMed, ProQuest, EBSCO, Clinicaltrials.gov, and the International Clinical Trials Registry Platform (ICTRP)) accessed on 8 July 2022. There were 11,938 electronic database documents discovered in all. A total of 11,871 articles were removed from the study due to being irrelevant (title, conference, published, editorial, news, comments, and reviews). In addition, 11 articles were eliminated from the study because they had a title that was duplicated across four search engines. Following that, 16 papers were eliminated since the abstracts did not meet the inclusion requirements. A total of 41 full articles were reviewed for eligibility and 25 were included in the qualitative synthesis (Figure 1).

### 3.2. Characteristics of the Studies

This systematic review was comprised of 19 studies with randomized trial designs [36,37,38,39,40,41,42,43,44,45,46,47,48,49,50,51,52,53,54] and 6 studies [55,56,57,58,59,60] with other types of trial designs (pragmatic trial and quasi-experimental). The features of the studies are detailed in Table 1. The characteristics of the participants and the mobile health intervention are summarized in Table 2. A total of 7600 participants from 19 trials were randomized, with 6318 completing the study, indicating that ±17% of the participants did not finish the randomized trials. A total 1866 (94.7%) completed the studies from non-randomized trial design. There are eight studies [38,47,49,50,56,58,59,60] that samples of less than 100 participants. There were three studies with a sample size of more than 1300, two RCT studies [52,61] and 1 pragmatic study [57]. The majority of studies reported mobile health as an early intervention for individuals with prediabetes. Chen’s [55] study reported not only personal counseling as an early intervention but also risk scoring as screening. 

A total of thirteen studies [36,39,40,41,43,44,46,48,50,51,53,57,58] only used hbA1c as a diagnostic tool for prediabetes. Research conducted by Ramachandran [36] only used an oral glucose tolerance test (OGTT) as a diagnostic standard for prediabetes. Three other studies, namely Chen [55], Block [37], and Muralidharan [54], only used plasma fasting glucose (FG) tests. There were seven other studies that use a combination examination to establish the diagnosis of prediabetes. Bender, Sharit, Kunthi, Stewart, Sevilla, and Moravcova [42,45,49,56,59,60] used three tests, namely HbA1c, OGGT, and Fasting Glucose (FG) levels. Meanwhile, Fukuoka [47] used a combination of HbA1c and FG assays. There was one study that diagnosed prediabetes by not using laboratory tests but using risk category scoring from the ADA [38].

The youngest average age of 37.8 ± 9.2 years was found in Muralidharan’s [54] study, while the oldest average age was found in McLeod’s [53] study, which is 61.8 ± 9.5 years old. Hamdan’s [41] study only examined female subjects and Ramachandran’s [36] only examined male subjects. Bender’s study only intervened in Filipino Americans because they had a higher risk of developing type 2 diabetes [49]. The shortest studies lasted 3 months, and were found in the Sevilla, Kondo, and Griauzde [46,50,60] studies. Khunti’s study had the longest intervention which was 48 months. There were three types of interventions used, namely text message-based, website-based, and mobile phone apps. 

Staite [39] used all three intervention combinations. Block [37] used websites and mobile phone apps. Stewart [59] used text message-based and mobile phone apps. There were 15 studies using only mobile phone apps, 3 other studies using websites, and 4 studies using text messages only. There were 6 studies that use wearable devices as part of the intervention [39,43,49,50,59]. The study outcomes were classified into five categories, which included the incidence of diabetes mellitus (DM), anthropometric measures, laboratory examinations, measures of physical activity, and eating behavior. Ramachandran [36], Khunti [42], and Nanditha [52] examined all of the five outcome categories. Kondo and Chen examined four of the five categories except incidence of DM. Meanwhile Fischer [57] (anthropometric measures) and Francis [40] (physical activity measures) only assessed one of the five clinical outcome parameters.

### 3.3. Quality Assessment

The Jadad score [33,62] was used to measure the quality of the trials, which reflects the quality of research based on their description of randomization, blinding, and dropouts (withdrawals). The methodology features of the Jadad score are shown in Table 3. Each study in this systematic review received a score (Table 4). In the Jadad score, the scale runs from 0 to 5, with a score of ≤2 indicating a low-quality report and a score of ≥3 indicating a high-quality report [33]. Six [55,56,57,58,59,60] studies have low quality, according to the analysis. This is due to the trial’s insufficiency of randomization in selecting participants. Concerning “blinding,” we understand that many studies do not receive points because the nature of the intervention does not allow for “blinding”.

**Table 1 ijerph-19-13638-t001:** Characteristics of the studies.

Author, Year	Subject Criteria	Number of Subjects	Duration of Intervention and Study Location	Type of Mobile Health	Functions Used	Methods	Outcome Measures	Measurement Methods Detail
**Ramachandran et al., 2013 [36]**	**Impaired glucose tolerance** **(OGTT: 140–199 mg/dL)** **Age: 35–55 years** **BMI: ≥23 kg/m² or** **more.** **Sex: Male**	**Randomized Participants: 537** **Analyzed Participants:** **537** **Completed:** **527**	**24 months** **Southeast India**	**Text message based:** **Short message** **Service (SMS),** **Text Message** **based on the transtheoretical model stage**	**Encouraged lifestyle change could reduce incident type 2 diabetes**	**RCT** **(prospective,2-arm parallel-group)**	Primary Outcome Incident type 2 diabetes Secondary Outcome b. Body Mass Index (BMI) c. Waist circumference d. Blood Pressure (systolic and diastolic) e. Physical Activity score f. Total dietary intake g. Lipid Profile (total cholesterol, Triglyceride (TG), HDL) h. HOMA-IR	**Primary Outcome** **Incident T2DM**: The estimated cumulative incidence of T2DM was calculated using unadjusted Cox regression analysis. **Secondary Outcome** Body Mass Index (BMI), Waist circumference, and Blood Pressure (systolic and diastolic): Mean of two readings obtained at each visit using standardized methods. Physical activity evaluation using a questionnaire Total dietary intake: dietary intake as determined by a 24-h recall Lipid profile (total cholesterol, triglycerides (TG), and HDL): fasting venous plasma sampling HOMA-IR: ([fasting insulin (mU/L) × fasting glucose (mmol/L)] formula/22·5)
**Chen et al., 2014 [55]**	Impaired Fasting Glucose (FG:100–125 mg/dL) Age: 40–70 years BMI: Not Reported Sex: Male and Female	**Eligibility Participant**: 253 Analyzed Participant: 231 Completed: 231	6 months Lu’an Anhui, China	Website Based: SWAP-DM2	Risk scoring and personalized counseling lifestyle management	**Quasi-Experimental** **Mixed Methods**	Change mean in weight Change mean BMI Change mean BP (Systolic, Diastolic) Change mean Vegetable Intake Calorie Intake Modifying Diet Physical Activities Leisure-Time Exercises Engaging Relatives in diabetes prevention	not reported in detail
**Block et al., 2015 [37]**	Impaired Fasting Glucose (FG: 100–125 mg/dL) and or Prediabetes (A1c: 5.7–6.4%) Age: 30–69 years BMI: ≥ 27 kg/m (BMI ≥25 kg/m for Asian participants) Sex: Male and Female	Randomized Participants: 340 Analyzed Participants: 339 Completed: 292	6 months Berkeley USA	“Hybrid” Mobile phone apps and Website Based: The Alive-PD	To improve glycemic control and reduce diabetes risk through lasting changes in physical activity and eating habits	RCT (wait-list controlled Trial)	Primary Outcome change in mean HbA1c change in mean Fasting Glucose Secondary Outcome change in mean weight change in mean BMI change in mean waist circumference, change in mean triglyceride (TG) to high-density lipoprotein cholesterol (HDL-C) ratio (a proxy measure for insulin resistance)	not reported in detail
**Fukuoka et al., 2015 [38]**	Impaired Fasting Glucose (FG: 100–125 mg/dL) and or Prediabetes (A1c: 5.7–6.4%) Age: ≥35 years BMI: BMI of at least 25 kg/m (BMI > 23 kg/m for Asian-Pacific Islanders participants) Sex: Male and Female	Randomized Participants: 61 Analyzed Participant: 61 Completed: 56	5 months San Francisco and Berkeley, CA, USA	Mobile phone apps based: The mDPP trial app	to supplement the in-person sessions and was intended to enhance their effect. Electronic diaries for self-monitoring of weight, activity, and caloric intake, daily reminders.	RCT (2-arm, parallel groups)	Primary Outcome change in mean Weight change in mean BMI Secondary Outcome Hip circumference Glucose levels Physical Activity Lipid profile Blood pressure	All outcomes are objectively measured and reported by participants. but not in detail
**Fischer et al., 2016 [39]**	Prediabetes (HbA1c from 5.7–6.4%) and Obesity (BMI 25–50 kg/m^2^) Age: ≥18 years BMI: 25–50 kg/m^2^ Sex: Male and Female	Randomized Participants: 163 Analyzed Participants: 157 Completed Participants: 157	12 months Denver, Colorado, USA	Text message based: Text message using the National DPP (Diabetes Prevention Program) curriculum content (SMS4PreDM)	To support weight loss	RCT (parallel groups)	Primary Outcome change in mean weight Secondary Outcome Achieving ≥3% weight loss Achieving ≥5% weight loss Change in mean HbA1c Change in mean Systolic Blood pressure Operating costs per participant receiving the intervention	not reported in detail
**Bender et al., 2018 [40]**	Prediabetes (HbA1c from >5.6%) or Impaired Fasting Glucose (FG; 100–125 mg/dL) Or Impaired glucose tolerance (OGTT: 140–199 mg/dL) Age: ≥18 years BMI: >23 kg/m^2^ Sex: Male and Female Notes: Self-identified as Filipino	Randomized Participants: 67 Analyzed Participants: 61 Completed Participants: 61	6 months San Francisco, USA	Mobile phone apps based: (Fit and Trim) App Plus Wearable Devices (Fitbit Devices)	Weight loss lifestyle intervention including virtual social support	RCT (2-arm, wait-list controlled Trial)	Primary outcome: Feasibility, Engagement Retention Secondary outcome: Achieving ≥ 5% weight loss Change in mean Waist circumference Change in mean Fasting plasma glucose Change in mean HbA1c Change in mean BMI	not reported in detail
**Sharit et al., 2018 [56]**	Prediabetes (HbA1c from 5.7–6.4%) or Impaired Fasting Glucose (FG: 100–125 mg/dL) Or Impaired glucose tolerance (OGTT: 140–199 mg/dL) Age: ≥20 years BMI: 25–42 kg/m^2^ Sex: Male and Female	**Eligibility Participant**: 38 Analyzed Participants: 38 Completed Participants: 38	13 weeks A large city in the USA	Website based: Practice on use of Track Health (TH) Journals and Vitals + Readings features as a basis for promoting positive Physical Activity (PA) and dietary lifestyles	Promoting positive Physical Activity (PA) and dietary lifestyles	Quasi-Experimental (pilot 3 month clinical trial pre-post design)	Primary outcome: Weight BMI Abdominal circumference Physical activity, Dietary Intake BP (systolic and diastolic) blood pressure Secondary outcome: Exercise self-efficacy, Diet self-efficacy, Intent to perform, Intent to adhere to diet, Patient activation	not reported in detail
**Griauzde et al., 2019 [41]**	Prediabetes (HbA1c from 5.7–6.4%) Age: >18 years old BMI: Not Reported Sex: Male and Female	Randomized Participants: 69 Analyzed Participants: 55 Completed Participants: 55	12 weeks Ann Arbor, MI, USA	Mobile phone apps based: App-Only And App-Plus Wearable device (Fitbit Devices)	To help individuals gain awareness of and control over the factors that influence their health behaviors (Sleep; Presence; Activity; Creativity; Eating) (S.P.A.C.E).	RCT (parallel, 3-arm) **Mixed Methods**	a. Primary Quantitative Measures: - Feasibility and - Acceptability of mobile health apps b. Secondary Quantitative Measures: - autonomous motivation to prevent T2DM c. Qualitative Measures: - Participants’ experiences with the app and Fitbit devices and behavioral changes that occurred	not reported in detail
**Fischer et al., 2019 [57]**	Prediabetes (HbA1c from 5.7 06.4%) Age: > 18 years old BMI: Not Reported	Eligibility Participant: 1.518 Analyzed Participants: 1.518 Completed Participants: 1.492	12 months Denver, Colorado, USA	Text message based: Text message using the National DPP (Diabetes Prevention Program) curriculum content (SMS4PreDM)	To support weight loss	**Pragmatic trial**	Weight change Achieving ≥3% weight loss SMS4PreDM delivery costs	For SMS4PreDM participants, a repeating measure analysis used all weights available in the EHR from routine healthcare visits within a year of the individual’s start date, or matched identification dates for controls.
**Ramos et al., 2020 [42]**	Prediabetes (HbA1c from 5.7–6.4% within 3 months before study enrollment) Age: >18 years old BMI: NA Sex: Male and Female Notes: referral from the patient’s physician	Eligibility Participant: 1.513 Randomized Participants: 202 Analyzed Participants: 202 Completed Participants: 155	12 months New York, NY, USA	Mobile phone apps based: Noom’s App With the National DPP (Diabetes Prevention Program) curriculum content	To support weight loss	RCT (two-arm, parallel RCT)	Primary outcome: Change in Weight Change in HbA1C Level Secondary outcome: Program engagement Numbers of logged meals, Numbers of logged weight, Numbers of logged steps, Numbers of articles read, Numbers of posts in the group, Numberers of messages to the coach	The DCA Vantage (Siemens) point-of-care (POC) HbA1c machine was used in primary care and endocrinology clinics.
**Nandhita et al., 2020 [43]**	Prediabetes (HbA1c from 6.0–6.4%) Age: 35–55 years (India) 40–74 years (UK) BMI: ≥23 kg/m^2^ (India) Sex: Male and Female Notes: Pre-screening: having three or more risk factors, including age 35–55 years, BMI ≥ 23 kg/m2, waist circumference ≥90 cm in men and ≥80 cm in women, first-degree family history of type 2 diabetes, history of hypertension or prediabetes, or habitual sedentary behavior	Randomized Participants: 2062 Analyzed Participants: 2062 Completed Participants: 1.763	24 months UK and India	Text message based: Short message Service (SMS), Text Message based on the transtheoretical model stage	To provide additional education and motivation. The messages provided tips, suggestions, and positive reinforcement for healthy behaviors including goal setting, physical activity, dietary planning, and personal strategies for a lifestyle change.	RCT (two-arm, parallel RCT)	Primary outcome: Incident type 2 diabetes by HbA1c international criteria for fasting plasma glucose or HbA1c at any study review visit or in any healthcare setting (UK) b. Incident type 2 diabetes by HbA1c alone (India) secondary outcomes: Body Weight Body Mass Index (BMI) Waist circumference Blood Pressure (systolic and diastolic) Fasting Blood Glucose levels Lipid levels Proportion achieving HbA1C ≤ 6% Acceptability of SMS Dietary variables, Physical Activity score Quality of life	HbA1c international criteria for fasting plasma glucose or HbA1c at any study review visit or in any healthcare setting (UK); incident type 2 diabetes determined solely by HbA1c (India) Body Mass Index (BMI), Waist Circumference, and Blood Pressure (Systolic and Diastolic): the average of two readings taken by standard procedures at each visit. The serum lipid profile (total cholesterol, low-density lipoprotein, HDL-cholesterol, and triacylglycerols) and HbA1c were measured using standard enzymatic procedures with quality control.
**Mcleod et al., 2020 [44]**	Prediabetes (HbA1c from 5.9–6.6% current or tested in preceding 3 months) Age: 18–75 years BMI: ≥20–40 kg/m^2^ Sex: Male and Female Notes: **participant included diabetes**	Randomized Participants: 225 Analyzed Participants: 201 Completed Participants: 201	12 month The greater Wellington and Waikato regions of the North Island of New Zealand,	Mobile phone apps based: BetaMe/Melon App	a structured, in-person lifestyle modification program focused on weight reduction	RCT (parallel-group 2-arm single-blinded superiority trial)	Primary outcome: Change in HbA1C Change in Weight Secondary outcomes: c. BMI d. Waist circumference e. systolic and diastolic BP	The Protocol describes standardized measurement procedures.
**Muralidharan et al., 2020 [45]**	Prediabetes Impaired Fasting Glucose (FG 100–125 mg/dL) Age: 20–65 years BMI: ≥25 kg/m^2^ Sex: Male andFemale	Randomized Participants: 741 Analyzed Participants: 561 Completed Participants: 561	4 month Chennai, Bangalore and New Delhi, India.	Mobile phone apps based: mDiab	To support Diabetes prevention program (DPP) and culturally modified it to suit the Indian population	RCT (parallel-group two-arm)	Waist circumference Body fat percent, Blood pressure (BP), Fasting Plasma glucose Serum lipid (triglycerides, total cholesterol)	Omron HBF-306 Body Fat Monitor was used to calculate the percentage of body fat. Every day, the waist circumference was measured with a standard, non-stretchable, 1-inch tape measure that was calibrated. While calibrating, it was ensured that the difference between the two readings was no greater than 0.2 cm. If the measuring tape was damaged, whether stretched or twisted, or if graduations had been erased, it was replaced with a new one. The study participants were instructed to stand with their feet together and their arms by their sides, palms facing inward. The technician then determined the midpoint by locating the inferior margin of the last rib and the crest of the ileum. After ensuring the proper placement of the tape around the waist, a measurement to the nearest 0.1 cm was taken. Two measurements were taken, and the average of the two was taken.
**Xu et al., 2020 [46]**	**Prediabetes** **(High risk for diabetes,** **measured by the American Diabetes Association (ADA)** **screening tool (score of ≥5 or)** Age: ≥18 years BMI: NA Sex: Male and Female Notes: access to WeChat	Randomized Participants: 81 Analyzed Participants: 76 Completed Participants: 76	6 month Beijing, China	Mobile phone apps based: DHealthBar	improving eating habits and physical activity,	RCT (A pragmatic, parallel-group, 2-arm)	Primary Outcome: one’s intention for behavior change, (in dietary and physical activity) Secondary Outcome: stage of change for dietary behaviors and physical activity Waist circumference BMI	Participants filled out self-reported questionnaires.
**Staite et al., 2020 [47]**	Prediabetes (HbA1c from 5.7–6.4%) Age: 18–65 years BMI: ≥25 kg/m² (≥23 kg/m² if of Asian ethnicity) Sex: Male and Female Notes: being ambulatory	Randomized Participants: 200 Analyzed Participants: 156 Completed Participants: 156	12 month London, United Kingdom.	“Tribrid” Mobile phone Apps based & Web-based & Text Messages based: the wristband and the associated study-specific smartphone app (Buddi wristband)	to support participants in forming healthy intentions, encourage self-monitoring of lifestyle behaviors, and promoting social support	RCT (two-arm, parallel, single-blind)	Primary outcome: Change in weight Secondary outcome: Change in HbA1c levels (categorical) Blood Pressure (BP) Waist to hip ratio Lipid levels	Using a stadiometer, weight was measured in light clothing, without shoes, to 0.01 kg, and height was measured to 0.1 cm (Class 3 Tanita SC240). BMI (kg/m^2^) was calculated based on weight and height measurements. Using a non-extensible steel tape against the bare abdomen, the waist circumference (cm) was measured horizontally halfway between the lowest rib and the upper prominence of the pelvis. The waist-to-hip ratio was also calculated by measuring hip circumference. Diastolic and systolic blood pressure (BP), as well as resting heart rate, were measured using digital Omron BP monitors (Omron M7) and standardized procedures for the average of two readings taken one minute apart while seated.
**Francis et al., 2021 [48]**	Prediabetes (HbA1c from 5.7–6.4%) And ≥25 kg/m^2^ Or Person with ≥30 kg/m^2^ Only Age: >18 years old BMI: ≥25 kg/m^2^ Sex: Male and Female	Randomized Participants: 430 Analyzed Participants: 388 Completed Participants: 388	6 months Iowa City, IA, USA	Mobile phone apps based: MapTrek-Plus Fitbit Devices	To promotes walking	RCT (two-arm, parallel)	Primary outcome: total number of daily steps as objectively measured by the Fitbit Zip activity monitor Secondary outcome: number of minutes per day with at least 100 steps active minutes.	The Fitbit Zip activity monitor objectively measures the number of daily steps.
**Hamdan et al., 2021 [49]**	Prediabetes (HbA1c from 5.6–6.9%) Age: 18–60 years BMI: ≥25 kg/m^2^ Sex: Female Notes: active users of android or IOS-based	Cluster Randomized: 3 primary care Randomized Participants:110 Analyzed Participants: 110 Completed Participants: 483	6 months Riyadh, SA	Mobile phone apps based: Al-Nahdi Mobile App And Social Media	Lifestyle modifications emphasizing the importance of weight loss, healthy diet and physical activity.	RCT (multicenter, 3-arm cluster randomized, Multi-intervention)	Primary outcome: Change in HbA1c Change in weight Secondary outcome: Blood lipid Dietary change	Anthropometric measurements were taken at baseline and after 6 months in all three arms of the study. Height (to the nearest cm) and weight (to the nearest 100 g) were measured in light clothing without shoes on calibrated scales. Waist circumference (cm) was measured at the umbilical level without clothing after exhaling in a relaxed standing position. The traditional mercurial sphygmomanometer was used to measure blood pressure (mmHg) after a sufficient rest at each visit, and the average was recorded. Body mass index (BMI) is calculated by dividing weight in kg by height in meters squared (kg/m^2^). All measurements were taken by trained and licensed nurses and dietitians.
**Summers et al., 2021 [58]**	Prediabetes (HbA1c from 5.6–6.9%) Age: ≥18 years BMI: Not reported Sex: Male and Female Notes: Diabetes and prediabetes	**Eligibility Participant: 27** Analyzed Participants: 27 Completed Participants: 21	12 months Norwood Surgery in Southport, United Kingdom.	Mobile phone apps based: The Low Carb Program App	Educates and supports sustainable dietary changes focused on carbohydrate restriction	**Single-Arm Prospective Study**	Primary outcome: Change in HbA1c Change in weight	not reported in detail
**Khunti et al., 2021 [50]**	Prediabetes (HbA1c from 6–6.4% within the last 5 years.) or Impaired Fasting Glucose (FG 100–125 mg/dL) Or Impaired glucose tolerance (OGTT: 140–199 mg/dL) Age: 40–74 years, or 25–74 years if they were South Asian BMI: Not Reported Sex: Male and Female	Randomized Participants: 1366 Analyzed Participants: 1366 Completed Participants: 986	48 month Leicester, United Kingdom.	Mobile phone apps based: an integrated mobile health	to support the maintenance of behavior change within Walking Away	RCT (three-arm, parallel-group, pragmatic, superiority)	Primary outcome: Change ambulatory activity (steps per day) at 48 months Secondary outcome: time spent sedentary in light and moderate to vigorous-intensity physical activity time spent in the postures of sitting/lying, standing, and walking Physical Activity Questionnaire Biochemical variables (comprising HbA1c, lipid profile (triglycerides, HDL, LDL and total cholesterol), urea and electrolytes (sodium, potassium, urea and creatinine), and liver function tests (albumin, total bilirubin, alkaline phosphatase and alanine transaminase). e. Standard anthropometric and demographic measurements f. Genetics g. Cardiovascular risk h. Sleep i. Self-reported dietary behavior j. Health-related quality of life	Estimated conversion rate at the lower level reported for traditionally defined prediabetes Height, body weight, body fat percentage, and waist circumference were all measured to the nearest 0.5 cm, 0.1 kg, 0.5%, and 0.1 cm, respectively. The waist circumference was measured with a soft tape measure halfway between the lowest rib and the iliac crest. When the participant was seated, the arterial blood pressure was taken from the right arm. Venous sampling was used to assess standard biomedical outcomes such as HbA1c, a lipid profile (triglycerides, HDL, LDL, and total cholesterol), urea and electrolytes (sodium, potassium, urea, and creatinine), and liver function tests (albumin, total bilirubin, alkaline phosphatase, and alanine transaminase).
**Katula et al., 2022 [51]**	Prediabetes (HbA1c from 5.7–6.4%) Age: ≥19 years BMI: ≥25 kg/m^2^ (≥22 kg/m^2^ if participant self-identified as Asian) Sex: Male and Female	Randomized Participants: 599 Analyzed Participants: 599 Completed Participants: 483	12 months Omaha, Nebraska., USA	Mobile phone apps based: App-Plus Wearable Devices	To support weight loss	RCT (two-rm, parallel single-blind)	Primary outcome: change in HbA1c Secondary outcome: Change in weight The proportion of participants that lost ≥ 5% of initial weight Improvement in the diabetes risk categories Blood lipid Blood pressure	All study measures were collected at the University of Nebraska Medical Center in a 4-week assessment window by staff who were blinded to study group assignment at each assessment point (baseline, 4 months, and 12 months). Sociodemographic data and health literacy were only collected at the start of the study. HbA1c levels were determined using nonfasting blood samples. Blood was drawn via venipuncture and processed at the University of Nebraska Medical Center’s central diagnostic testing laboratory for cardiovascular disease (lipid panels) using the boronate affinity analytical technique. Weight was measured in stocking feet on a calibrated medical-grade scale with the participant fasting.
**Stewart et al., 2022 [59]**	Prediabetes (HbA1c from 5.7–6.4%) Age: Not Reported BMI: Not Reported Sex: Male and Female Notes: One method of participant recruitment from primary care manager (PCM) referral	Eligibility Participant: 33 Analyzed Participants: 33	12 month Augusta, Georgia, USA	“Hybrid” Mobile phone apps and text message-based: Mobile phone App Plus Wearable Devices (FitBit) And The combination of the daily text messages of DPP content	to support participants in losing weight (5–7%) and being active (150 physical activity minutes/week)	**Quasi-experimental** **(pre/post design)**	Change in weight, Physical activity, Sedentary time.	Baseline biometric/survey measures and the electronic medical record Physical activity as self-reported
**Lim et al., 2022 [52]**	Prediabetes (HbA1c from 6–6.4%) or Impaired Fasting Glucose (fasting glucose 100–125 mg/dL) Or Impaired glucose tolerance (oral glucose tolerance test: 140–199 mg/dL) Age: 21–75 years BMI: ≥23 kg/m^2^ Sex: Male and Female	Participant Eligible: 217 Randomized Participants: 148 Analyzed Participants: 148 Completed Participants: 140	6 month Singapore	Mobile phone apps based: the nBuddy Diabetes app	To empower individuals through prompts and cues and achieve clinically meaningful weight loss of >5%	RCT (multicenter concurrent parallel RCTs)	Primary outcome: mean weight loss, physical activity, sedentary time. Secondary outcome: mean changes in HbA1c, fasting blood glucose (FBG), blood pressure, serum lipids, creatinine, dietary intake, physical activity	Body weight was measured in the clinic by research staff using a standard digital weighing scale (Omron HN-289, Japan) after an overnight fast, with participants dressed lightly and without shoes. To calculate BMI, height was measured without shoes to the nearest centimeter. Venous blood samples were collected after 8–12 h of overnight fasting and processed at CAP-accredited laboratories (National University Hospital Department of Laboratory Medicine or National Healthcare Group Diagnostics). Plasma glucose was determined using a photometric assay using the hexokinase method, and HbA1c was measured using high-performance liquid chromatography. An enzymatic colorimetric assay was used to determine serum lipids and creatinine levels. Self-reported questionnaires were used to collect participants’ physical activity levels in minutes per week at baseline, 3 months, and 6 months. At the baseline, 3-, and 6-month visits, dietary intake was collected using a 2-day food diary and analyzed using the nBuddy Dashboard’s nutrient analysis platform, which includes over 14,000 food items and incorporates the Singapore energy and nutrient composition of food, Malaysian Food Composition, and USDA food databases, as well as nutritional information from food packaging and nutrient analysis of recipes.
**Sevilla et al., 2022 [60]**	Prediabetes (HbA1c from 6–6.4%) or Impaired Fasting Glucose (FG 100–125 mg/dL) Or Impaired glucose tolerance (OGTT: 140–199 mg/dL) Age: 18–65 years BMI: ≥20–40 kg/m^2^ Sex: Male and Female Notes: Some participants were randomly prescribed 750 mg extended release metformin every 12 h to evaluate the “medication” module of the web platform	**Eligibility Participant:** **122** Randomized Participants: 77 Analyzed Participants: 77 Completed Participants: 77	3 month Mexico City, Mexico	Web-based: Vida Sana	Lifestyle modification counseling with the goal of reaching a weight loss of >3%.	**Prospective Interventional Study**	The feasibility of Vida Sana changes in fasting glucose, glucose at 120 min, body fat percentage, waist circumference, visceral adipose tissue, free-fat mass index	Dual-energy X-ray absorptiometry is used to determine body composition. Waist and hip circumferences (to the nearest 0.5 cm) were measured at the midpoint between the lower ribs and the iliac crest and at the level of the trochanter major, respectively. Participants who met the inclusion criteria were invited back the following week for an intervention visit. Using colorimetric enzymatic methods, measurements from the oral glucose tolerance test included glucose levels, insulin levels, and lipid profiles (Unicel DxC 600). Beckman Coulter Synchron Clinical System. A chemiluminescence assay was used to measure insulin levels (Access 2, Beckman Coulter). HbA1c was measured using a Variant II Turbo system (BIORAD) and a 4-mL peripheral blood sample was drawn via venipuncture using the standardized technique. Diet and physical activity questionnaires were administered.
**Moravcova et al., 2022 [53]**	Prediabetes (HbA1c from 6–6.4%) or Impaired Fasting Glucose (FG 100–125 mg/dL) Or Impaired glucose tolerance (OGTT: 140–199 mg/dL) Or Insulin resistance (IR) (HOMA-IR > 2.7) Age: 18–60 years BMI: ≥30 kg/m^2^ Sex: Male and Female Notes: **participant included diabetes**	Randomized Participants: 100 Analyzed Participants: 78 (3 months) 51 (6 months) Completed Participants: 78 (3 months) 51 (6 months)	6 month Mexico City, Mexico	Mobile phone apps based: Vitadio application	to provide individualized support in lifestyle modification and self-management	RCT (prospective, double-armed)	primary outcome: the effect of Vitadio on weight reduction when compared to an in-person program secondary outcome: b. Body composition, c. Waist circumference d. Glucose metabolism e. Lipid metabolism f. Liver function Notes: only the 3-month laboratory measurement is evaluated in this paper. The 6-month attrition rate was higher dropout were associated with the COVID-19 pandemic.	The study procedure includes four visits: at baseline, three months later, six months later, and twelve months later, with anthropometric and laboratory examinations performed at each visit. Blood samples were collected after a 12- to 14-h fast. A glucose hexokinase method was used to measure glucose. An enzymatic colorimetric test was used to examine lipids. A two-step sandwich enzyme immunoassay using monoclonal antibodies was used to measure serum insulin. Matthews’ formula was used to calculate HOMA-IR. Body composition was determined using bioelectrical impedance analysis with the InBody 370 and 15 impedance measurements at 5 body segments, as well as a tetrapolar 8-point tactile electrode system.
**Kondo et al., 2022 [54]**	Prediabetes Impaired Fasting Glucose (FG: 100–125 mg/dL) Age: 40–75 years BMI: ≥25 kg/m^2^ Sex: Male and Female Notes: participant has metabolic syndrome	Randomized Participants: 122 Analyzed Participants: 75 Completed Participants: 74	3 months Tokyo, Japan.	Mobile phone apps based: DialBeticsLite	Daily recording of several physical parameters, in addition to tracking lifestyle behavior, that is, diet and exercise	RCT (open-label, 2-arm, parallel-design	Primary Outcome: change in VFA (visceral fat and abdominal) Secondary Outcome: Waist circumference BMI Body weight Blood pressure (BP) Diet Exercise	VFA was calculated by distinguishing visceral fat from abdominal subcutaneous fat using current flow from two different routes (DU-ALSCAN, HDS-2000, Fukuda Colin). The HDS-2000 underestimates VFA when compared to computed tomography scans, but the correlation was very strong (r = 0.89). Because of its simplicity and noninvasiveness, the HDS-2000 can be a good option for evaluating VFA. To avoid variation in measurement procedures across raters, the VFA of all participants was measured by the same individual with sufficient experience. The person was able to see again after the group intervened. Secondary outcomes included changes in physical and metabolic parameters from baseline to the 3-month follow-up. Physical parameters included BW, WC, BMI, and BP. Blood tests were used to determine metabolic parameters such as cholesterol, triglyceride, fasting plasma glucose, and HbA1c levels.

**Table 2 ijerph-19-13638-t002:** Summary of participant characteristics and mobile health intervention.

**Results**			Chen et al., 2014 [55]	Fischer et al., 2019 [57]	Sevilla et al., 2022 [60]	Sharit et al., 2018 [56]	Stewart et al., 2022 [59]	Summers et al., 2021 [58]	Bender et al., 2019 [40]	Block et al., 2015 [37]	Fischer et al., 2016 [39]	Francis et al., 2021 [48]	Fukuoka et al., 2015 [38]	Griauzde et al., 2019 [41]	Hamdan et al., 2021 [49]	Katula et al., 2022 [51]	Khunti et al., 2021 [50]	Kondo et al., 2022 [54]	Lim et al., 2022 [52]	Mcleod et al., 2020 [44]	Moravcova et al., 2022 [53]	Muralidharan et al., 2020 [45]	Nandhita et al., 2020 [43]	Ramachandran et al., 2013 [36]	Ramos et al., 2020 [42]	Staite et al., 2020 [47]	Xu et al., 2020 [46]
**Type of Trial**			Non-Randomized Intervention Trial						Randomized Trial																		
**Number of Samples Included**	**<100 participant**				77	38	33	27	67				61	69													90
	**100–500 participant**		253							340	163	430			253			122	148	225	100				202	208	
	**501–1000 participant**															599						833		537			
	**>1000 participant**			1518													1366						2062				
**SEX**			Both	Both	Both	Both	Both	Both	Both	Both	Both	Both	Both	Both	**Female**	Both	Both	Both	Both	Both	Both	Both	Both	Male	Both	Both	Both
**Mean AGE (years) ****			Range: 50–70	IG: 45.5 (12.2); CG: 48.4 (14.6)	IG: 48 ± 12.0; CG: 48.4 ± 10.8	57.7 ± 7.7	44 ± 8.5	52.42 ± 13.43	IG: 42.1 ± 12.2; CG: 41.3 ± 12.1	55 ± 8.9 range: 31–70	IG: 47.7 ± 12.4; CG: 45.2 ± 10.6	IG: 46.9 ± 13.2; CG: 45.8 ± 13.8	55.2 ± 9.0	IG: 52.1 ± 12.0; IG- plus 51.6 ± 11.1; CG: 51.3 ± 11.0	IG: 43.7 8.1 CG positive: 42.9–12.2: CG negative: 50.9–7.1	IG: 55.3 ± 12.9; CG: 55.6 ± 12.6	IG: 59.3 ± 9.1; CG positive: 59.4 ± 9.4; CG negative: 59.4 ± 8.8	IG: 49.3 ± 6.1; CG: 48.5 (5.3)	IG: 51.9 ± 8.7; CG: 54.3 (9.9)	IG: 61.8 ± 9.5; CG: 62.4 ± 8.7	IG: 43.3 ± 10.5; CG: 43.3 ± 8.4	IG: 37.8 ± 9.2; CG: 37.8 ± 9.6	CG: 52.0 ± 10.2; IG: 52.1 ± 10.3	IG: 46.1 ± 4.6; CG: 45.9 ± 4.8	IG: 55.7; CG: 57.5	IG: 51.76 ± 7.68; CG: 52.78 ± 8.20	IG: 46.0; CG: 47.5
**Mean AGE (years) ****		MIDDLE AGE (40–60)	√	√	√	√	√	√	√	√	√	√	√	√	√	√	√	√	√		√	√	√	√	√	√	√
		ELDERLY (>60)	√							√										√							
**Mean BMI (kg/m^2^) ****			IG: 24.80 ± 3.21; CG: 23.36 ± 2.95	Not Reported	IG: 30.8 ± 16.5 CG: 30.6 ± 4.2	33.6 ± 3.9	Pre:38.8 ± 1.8; Post: 37.6 ± 1.9	Not Reported	IG30.5(±3.9) CG 30.5(±4.9)	31.1 ± 4.4	Not Reported	IG: 36.4 ± 6.2; CG: 37.0 ± 6.8	33.3 ± 6.0	IG only: 33.0 ± 10.4; IG plus 30.7 ± 9.3; CG: 33.4 ± 7.8	IG: 30 ± 5.1; CG Positive: 34.8 ± 9.0; CG Negative: 31.6 ± 5.8	IG: 35.8 ± 6.1; CG: 35.8 ± 6.1	IG: 28.4 CG Positive: 28.2; CG negative: 28.5	IG: 27.4 3.0Con: 26.6 (2.2)	IG: 29.8 ± 4.2; CG: 29.8 ± 3.9	IG: 33.5 ± 7.7; CG: 33.1 ± 7.1	IG: 40.5 ± 7.1 CG: 39.7 ± 5.1	IG: 29.4 ± 3.8; CG: 29.3 ± 4.2	IG: 28.7 ± 4.7CG: 28.9 ± 4.8	IG 25.8 ± 3.0; CG: 25.8 ± 3.3	IG: 31.25 ± 6.43; CG: 30.94 ± 7.23	Not Reported	IG: 25.3; CG: 24.7
**Diagnostic Criteria**	**HbA1c**			√	√	√	√	√	√		√	√	√	√	√	√	√	√	√	√	√		√		√	√	
	**HbA1c Only**			√				√			√	√		√	√	√		√	√	√			√		√	√	
	**OGGT**				√	√	√		√								√				√			√			
	**OGGT Only**																							√			
	**FG**		√		√	√	√		√	√			√				√				√	√					
	**FG Only**		√							√												√					
	**Combination**				√	√	√		√				√				√				√						*
**Duration of study**	**<6 month**		√		√	√			√				√	√				√				√					
	**6–12 month**			√			√	√		√	√	√			√	√			√	√	√				√	√	√
	**13–24 month**																										
	**>24 month**																√						√	√			
**Type of mobile health**	**Text Message-Based**			√			√				√												√	√		√	
	**Web Based**		√		√	√				√																√	
	**Mobile Phone Apps**						√	√	√	√		√	√	√	√	√	√	√	√	√	√	√			√	√	√
	**Combination** **(Hybrid or Tribrid)**						√			√																√	
**Wearable** **Devices**			√					√		√		√												√	√		
**Outcome Measures**	**Incidence**																√						√	√			
	**Anthropometric Measures**		√	√	√	√	√	√	√	√	√		√		√	√	√	√	√	√	√	√	√	√	√	√	√
	**Laboratory Parameters**		√		√			√	√	√	√		√		√	√	√	√	√	√	√	√	√	√	√	√	
	**Physical Activity**		√			√	√				√	√	√				√	√	√				√	√			√
	**Dietary Behavior**		√		√										√		√	√					√	√			√

√: Reported. 
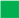
: Non-Randomized Interventional Trial. 
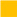
: Randomized Trial. *: Diagnosed by American Diabetes Association (ADA) screening tool (score of ≥ 5 or). **: Age (years) by mean ± SD or range age of participants. IG: Intervention Group; CG: Control Group.

**Table 3 ijerph-19-13638-t003:** Jadad Score Calculation [33,62].

METHODOLOGICAL FEATURE	ITEM	ANSWER	SCORE
**Randomization**	Was the study described as randomized?	Yes	1
		No	0
	Was the method used to generate the sequence of randomization described and appropriate?	Yes	1
		No	0
		Described and inappropriate	−1
**Blinding**	Was the study described as double-blind?	Yes	1
		No	0
	Was the method of double blinding described and appropriate?	Yes	1
		No	0
		Described and inappropriate	−1
**withdrawals and dropouts**	Was there a description of withdrawals and dropouts?	Yes	1
		No	0

**Table 4 ijerph-19-13638-t004:** Jadad quality score of experimental studies in this systematic review.

**Author, Year**	Ramachandran et al., 2013 [36]	Chen et al., 2014 [55]	Block et al., 2015 [37]	Fukuoka et al., 2015 [38]	Fischer et al., 2016 [39]	Bender et al., 2019 [40]	Sharit et al., 2018 [56]	Griauzde et al., 2019 [41]	Fischer et al., 2019 [57]	Ramos et al., 2020 [42]	Nandhita et al., 2020 [43]	Mcleod et al., 2020 [44]	Muralidharan et al., 2020 [45]	Xu et al., 2020 [46]	Staite et al., 2020 [47]	Francis et al., 2021 [48]	Hamdan et al., 2021 [49]	Summers et al., 2021 [58]	Khunti et al., 2021 [50]	Katula et al., 2022 [51]	Stewart et al., 2022 [59]	Lim et al., 2022 [52]	Sevilla et al., 2022 [60]	Moravcova et al., 2022 [53]	Kondo et al., 2022 [54]
**Randomization**	**1**	**0**	**1**	**1**	**1**	**1**	**0**	**1**	**0**	**1**	**1**	**1**	**1**	**1**	**1**	**1**	**1**	**0**	**1**	**1**	**0**	**1**	**0**	**1**	**1**
	**1**	**0**	**1**	**1**	**1**	**1**	**0**	**1**	**0**	**1**	**1**	**1**	**1**	**1**	**1**	**1**	**1**	**0**	**1**	**1**	**0**	**1**	**0**	**1**	**1**
**Blinding**	**0**	**0**	**1**	**1**	**0**	**1**	**0**	**0**	**0**	**1**	**1**	**1**	**1**	**1**	**1**	**1**	**1**	**0**	**1**	**1**	**0**	**1**	**0**	**1**	**1**
	**0**	**0**	**0**	**0**	**0**	**0**	**0**	**0**	**0**	**0**	**0**	**0**	**0**	**0**	**0**	**0**	**0**	**0**	**0**	**0**	**0**	**0**	**0**	**0**	**0**
**Withdrawals and Dropouts**	**1**	**1**	**1**	**1**	**1**	**1**	**1**	**1**	**1**	**1**	**1**	**1**	**1**	**1**	**1**	**1**	**1**	**1**	**1**	**1**	**1**	**1**	**1**	**1**	**1**
**Total Score**	3	1	4	4	3	4	1	3	1	4	4	4	4	4	4	4	4	1	4	4	1	4	1	4	4

Red: minimum randomization criteria score. Yellow indicates the lowest possible score for the blinding criteria. Green indicates the highest possible score for all criteria. Pink: 1 point off the total score.

### 3.4. Mobile Health Interventions

Six studies [36,39,48,52,57,59] used text messaging or short message service (SMS) as an intervention. Furthermore, Staite [39] and Stewart [59] include interventions other than text messaging, such as mobile phone apps and website-based interventions. Ramachandran [36] and Nanditha [52] reported on text messages based on the trans-theoretical model stage. Despite this, Staite [39] reported text messages that were based on the theory of planned behavior; Stewart [59] reported text messages that were based on Bandura’s self-efficacy: toward a unifying theory of behavioral change; and Fischer [48,57] did not report any details about the theory that underpins text messages to participants. Four studies [36,39,52,59] that used messages as interventions reported that content-text messages were created using curriculum material from the National Diabetes Prevention Program (DPP).

In a different manner, five studies [37,39,55,56,60] used website-based interventions. Besides website-based interventions, participants also received intervention via mobile phone apps [37]. Chen Study [55] with SOPs underlying SWAP-DM2 and Block Study [37] with DPP Curriculum demonstrate a distinct variation in the content of each website-based intervention in each study. The website-based intervention adopted the methods and theoretical frameworks, such as The SWAP-DM2 provided diabetes prevention services, ranging from uncomplicated educational websites and record-keeping to quite complex risk-scoring and individualized counseling [55]. Despite this, the Alive-PD was a fully automated and flexible online behavior change strategy [37]. The system includes tools for weight monitoring, eating, and physical activity, as well as weekly health information on diabetes and prevention strategies, quizzes, social support through virtual teams and a participant messaging program, feedback on diet and activity reports, and also goal achievement of success or failure, weekly reminders, and other features [37]. Practice the use of website-based was conducted as a basis for promoting positive Physical Activity (PA), dietary lifestyles [56] and mental resilience [39]. 

Eighteen [38,40,41,42,44,45,46,48,49,50,51,52,53,54,58] studies used mobile phone apps as an intervention. There were twelve mobile phone apps [37,40,42,45,46,47,51,52,53,54,59] developed from the Diabetes Prevention Programs (DPP) curriculum for adults at risk of T2DM. Despite this, another mobile phone app was developed by other methods such as for a low-carb program [58]. Most mobile phone apps aim to change dietary habits and increase physical activity. Reminder functions [37,38], daily recording [40,42,52], peer support [37,44], personal health coaching [44], and even gamified individual goals [48] were used to achieve the target parameters.

Six studies [40,41,47,48,51,59] were conducted on wearable devices, which may be useful for tracking and recording users’ activity in near-real time [47] and providing feedback [48]. Fitbit accounts for five out of six wearable devices [40,41,51,59,63].

### 3.5. Outcomes Reported

Anthropometric measures, such as weight loss, changes in BMI, and waist circumference were used as the primary outcomes in twelve studies [38,39,42,45,47,52,53,54,55,56,57,59]. Changes in HbA1c were the main result in five studies [37,44,49,51,58]. In addition, three studies [46,48,50] examined physical activity and its changes as the primary outcome of the research. In Ramachandran [36] and Nanditha’s [43] studies, the primary outcome was the incidence of type 2 diabetes. In the study conducted by Griauzde [41], Bender [40], and Sevilla [60], the primary outcomes were the feasibility and acceptability of the mobile health intervention. In Xu’s [46] studies, the primary outcome was a change in dietary behaviors and physical activity. The summary of intervention outcomes is shown in Table 5.

#### 3.5.1. Incidence of T2DM

The findings from Ramachandran [36] showed that mobile phone messaging (SMS) could be an effective technique for lifestyle modification to reduce the incidence of type 2 diabetes. The cumulative incidence of T2DM at 24-month follow-up was lower in those who received mobile phone messages than in controls [36]. Despite this, Khunti [50] and Nanditha [43] found that SMS and mobile phone apps did not significantly reduce the cumulative incidence of T2DM at the 12, 24, and 48-month follow-ups. There were other factors that could influence these inconsistent results, such as the content of the intervention and the role of the tool or other intervention (wearable device) [50] that has the potential to have an effect. Differences in examination methods and the provision of a prediabetes diagnosis were also factors that needed to be considered in these findings.

#### 3.5.2. Anthropometric Measures

An analysis of twenty studies using weight change as an outcome found that eight studies [39,43,44,47,49,50,53,60] showed no significant differences between intervention and control. Despite this, a significant difference was observed between the intervention and control groups in nine other studies [37,38,42,45,51,52,54,57,59]. Furthermore, the Sharit [56], Chen [55], and Summers [58] studies found significant changes in body weight following the intervention. Seven [37,38,42,45,51,57,59] of the 12 studies that found a significant difference in weight change from mobile health apps were based on content based on DPP.

There were two studies [39,57] that include a ≥3% weight loss as a research outcome. The results of the two are completely contradictory. According to the Fischer study in 2016, there was a significant difference in achieving ≥3% weight loss between the intervention and control groups [39], Meanwhile Fischer in 2019 [57] stated the opposite. Respondent characteristics, such as ethnicity, were suspected to be important considerations in these findings. There was a significant difference between the control and intervention groups and pre-post intervention in three [40,51,52] of the four studies that included achieving ≥5% weight loss as an outcome. Fischer [39], however, stated that there was no difference between the intervention and control groups. Two studies [40,51] used DPP-based content and mobile health apps as interventions.

Changes in BMI were noted in 17 studies [36,37,38,40,42,43,45,46,47,49,50,51,53,54,55,56,60]. BMI changes were shown in two studies [55,56] pre-and post-intervention, but the findings were not conclusive. The other fifteen studies [36,37,38,40,42,43,45,46,47,49,50,51,53,54,60] examined the comparison of changes in BMI in the intervention and control groups. Seven [37,38,42,45,46,51,54] of the fifteen studies reported a significant difference in BMI changes between the control and intervention groups. However, the findings of eight [36,40,43,47,49,50,53,60] other studies found, there was no significant difference in BMI changes between the intervention and control groups. 

Waist circumference was observed as an outcome in fifteen studies. Six studies [36,43,47,49,50,53] showed no significant effect of intervention on waist circumference. Despite this, eight studies [37,38,40,45,46,51,54,60] reported a significantly reduced waist circumference between the intervention group and the control. In addition, Sharit’s study described a different mean reduction in waist circumference between baseline and post-intervention [56]. 

**Table 5 ijerph-19-13638-t005:** Summary of Intervention Outcome.

**Results of Intervention Outcome**	Chen et al., 2014 [55]	Fischer et al., 2019 [57]	Sevilla et al., 2022 [60]	Sharit et al., 2018 [56]	Stewart et al., 2022 [59]	Summers et al., 2021 [58]	Bender et al., 2019 [40]	Block et al., 2015 [37]	Fischer et al., 2016 [39]	Francis et al., 2021 [48]	Fukuoka et al., 2015 [38]	Griauzde et al., 2019 [41]	Hamdan et al., 2021 [49]	Katula et al., 2022 [51]	Khunti et al., 2021 [50]	Kondo et al., 2022 [54]	Lim et al., 2022 [52]	Mcleod et al., 2020 [44]	Moravcova et al., 2022 [53]	Muralidharan et al., 2020 [45]	Nandhita et al., 2020 [43]	Ramachandran et al., 2013 [36]	Ramos et al., 2020 [42]	Staite et al., 2020 [47]	Xu et al., 2020 [46]
**Type of Trial**	**Non-Randomized Interventional Study**	Randomized Trial
**Incidence of type 2 DM**															**0**						**0**	**1**			
**anthropometry measures**	Weight Change	**1**	**1**	**0**	**1**	**1**	**1**		**1**	**0**		**1**		**0**	**1**	**0**	**1**	**0**	**0**	**0**	**1**	**0**		**1**	**0**	
Achieving ≥ 3% Weight loss		**0**							**1**																
Achieving ≥ 5%							**1**		**0**					**1**			**1**								
BMI Change	**1**		**0**	**0**			**0**	**1**			**1**		**0**	**1**	**0**	**1**			**0**	**1**	**0**	**0**	**1**	**0**	**1**
Waist circumference			**1**	**1**			**1**	**1**			**1**		**0**	**1**	**0**	**1**			**0**	**1**	**0**	**0**		**0**	**1**
**Physical Activity**	**1**			**0**	**1**					**1**	**0**			**1**	**0**	**1**	**0**				**0**	**0**			
**Dietary Behavior**	**1**		**0**										**0**		**0**	**1**					**1**	**1**			**1**
**Laboratory measures**	HbA1C Reduction			**0**			**1**	**0**	**1**	**0**		**0**		**0**	**1**	**0**	**0**	**1**	**0**	**0**		**0**		**0**	**0**	
Blood Glucose Level Reduction			**1**				**0**	**1**			**0**					**0**			**0**		**0**	**1**			
Fasting Plasma Glucose			**1**				**0**	**1**			**0**				**0**	**0**			**0**		**0**				
OGGT			**1**				**0**				**0**														
Blood Lipid Level Reduction								**0**			**0**		**0**				**0**			**0**	**0**	**0**			


: Not observed in the study. 
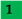
: Results were statistically significant. 
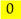
: Results were not statistically significant.

#### 3.5.3. Physical Activity 

Eleven [36,38,43,48,50,51,52,54,55,56,59] studies reported changes in physical activity as an outcome measure of the study. Five of the 11 studies stated that the mobile health intervention showed significant differences in increased physical activity between the control and intervention groups [48,51,54,59] and pre-post intervention [55]. Despite this, six studies reported that there were no significant differences in increases in physical activity between the control and intervention groups [36,38,43,50,52] and pre-post intervention [56]. 

#### 3.5.4. Dietary Behaviors 

Eight [36,43,46,49,50,54,55,60] studies reported dietary behavior as an outcome measure. Five [36,43,46,54,55] of the eight studies stated that the mobile health intervention showed significant differences in changes to healthy dietary behavior between the control group and the intervention and pre-post-intervention groups [55]. Despite this, three [49,50,60] studies reported that there was no significant difference in changes in dietary behavior between the control and intervention groups.

#### 3.5.5. Laboratory Measures

In sixteen studies [37,38,39,40,42,43,44,47,49,50,51,52,53,54,58,60], the HbA1c test was used as an outcome. Four studies found that mobile health interventions significantly improved HbA1c reduction, both in pre-post [51,58] studies and control intervention [37,52] studies. On the other hand, twelve [38,39,40,42,43,44,47,49,50,53,54,60] other studies found that there was no significant difference in HbA1c reduction between the control and intervention groups.

Reduction in blood glucose levels was one of the measures used as an outcome in eight studies [36,37,38,40,43,53,54,60]. Three [36,37,60] studies explained that there were significant differences in reduction in blood fasting glucose levels between the intervention and control groups and 1 h glucose levels in pre-post intervention. Whereas no significant difference in reduction in blood fasting glucose levels was found between the intervention and control groups in the other five studies [38,40,43,53,54]. In addition, the reduction in blood glucose levels was also significantly different in the oral glucose tolerance test (OGTT) in the control and intervention groups in the study on Vida Sana apps [60].

Seven studies [36,37,38,43,45,49,52] reported blood lipid level reduction as an outcome of the study. All of these studies stated that the mobile health intervention did not show a significant difference in blood lipid level reduction between the control and intervention groups [36,37,38,43,45,49,52].

## 4. Discussions 

Traditional face-to-face treatment for achieving and sustaining weight or BMI can be replaced with technologies that are practical, affordable, and scalable [64]. Mobile phone intervention has gained widespread acceptance among people of all ages and socioeconomic backgrounds, and it provides numerous opportunities in health care, including self-management and T2DM prevention [65]. Despite growing concerns about data privacy in the fields of access, patient confidentiality, and data storage [66]. The variety of definitions and use of diagnostic tools used in these study results indicate that prediabetes was a complex condition that triggers a burden on clinical services and public health policies [3]. The use of combined diagnostic tools is expected to be able to detect conditions of undiagnosed prediabetes and diabetes [67]. Depending on the diagnostic tools used as standards, the characteristics of the intervention each subject requires, and the treatment’s results can change. 

During long-term follow-up, there was no significant difference between mobile health interventions and controls in reducing the incidence of T2DM [43,50]. Despite this, there was variation in outcomes that may be brought about by variances in the intervention’s underlying theory and the message’s content to participants, and length of observation. Participants in twenty trials used technology to reduce weight, according to this systematic review. Studies show that mobile phone messaging is an effective and acceptable method to deliver advice and support towards dietary behavior to prevent T2DM in individuals at high risk. The findings [57] showed that although SMS4PreDM was relatively low-cost to deliver and demonstrated high retention, weight loss outcomes may not be sufficient to serve as a population health strategy. It can happen because pre-diabetes is influenced by many factors [55]. 

In these studies, website-based interventions were used to promote positive health outcomes such as physical activity, dietary habits, and mental resilience. These interventions differed in their complexity, ranging from simple educational websites to more complex individualized counseling. The differences in the details of the form of intervention, may be one factor making the research results inconclusive.

As a simple and effective measure of central obesity, waist circumference is a major predictor of increased risk of hypertension, diabetes mellitus, dyslipidemia, metabolic syndrome, and coronary heart disease [68]. Waist circumference could potentially be used as a clinical equivalent for visceral adipose tissue, which impairs insulin sensitivity and predisposes to prediabetes when excessive [69]. A systematic review and meta-analysis study looked at the impact of technology on waist circumference reduction and found a mean change of 02.99 cm (95% CI: 03.68 to 02.30) [68]. Self-management by chronic healthcare customers can be enhanced via mobile apps, according to users of health apps. This might be used for patients with prediabetes, particularly in terms of lifestyle adjustments [70].

Using text messages instead of other reminders had some advantages. Text messages could be sent to patients at the same time, they were always available, cost-effective, and required less staff [71]. Despite all the benefits and features of text messaging on a mobile phone, there are some drawbacks, such as: the staff could not be certain that text messages were received by all participants, mobile phone numbers could have changed [71], and participants sometimes blocked their status or stopped reading the text messages [72]. A few intervention strategies were bidirectional, allowing T2DM patients to receive continuous and personalized support via SMS while also allowing T2DM patients to interact with healthcare professionals and diabetes care educators to strengthen their T2DM self-management abilities and knowledge [65].

In other circumstances, mobile health technology (including social media) may provide benefits such as low or no cost, high scalability, self-tracking, and tailored feedback, image and video use for improved health literacy, wide range, and data sharing for large-scale analytics [73]. Mobile phone apps are widely utilized all around the world. Health and medical apps are increasingly being employed in a range of scenarios, according to evidence. Many authors in the medical and public health literature have highlighted the benefits that laypeople and healthcare professionals may contribute to health and medical apps [74]. Chronic disease or patients’ conditions must be monitored for a long time. Using mobile phone apps for chronic self-care could be beneficial, allowing patients to monitor and regulate their illnesses [70]. Despite the fact that mobile health intervention prompts were well-received by patients, there are conflicting results regarding their influence on behavioral, laboratory, and T2DM incidence [75].

There are some limitations in this study that need to be addressed. First, in our study, we only used American Diabetes Association (ADA) criteria for prediabetes, although many other prediabetes studies used the World Health Organization (WHO) criteria. Second, although few studies on mobile health interventions with a sample adequate for prediabetes intervention used randomized controlled trials (RCTs) as their study design, we propose that further studies on mobile health apps for prediabetes intervention use RCT design. Third, we were unable to depict all studies in meta-analysis due to differences in diagnostic methods, evaluation, and reporting of intervention outcomes. Approximately half of the studies provided sufficient data to compute them. Fourth, our search strategy may have resulted in a bias for positive results because null or negative results are less likely to be published. Finally, the full text of certain studies was not available.

## 5. Conclusions

The systematic review includes twenty-five studies that were discussed about mobile health intervention for pre-diabetes among middle-aged and elderly patients. A few studies showed that each mobile health intervention promised an effective and acceptable method to deliver advice and support towards lifestyle modification to prevent diabetes. Although, evidence of the effectiveness of mobile health as an intervention for prediabetes was inconclusive.

## Figures and Tables

**Figure 1 ijerph-19-13638-f001:**
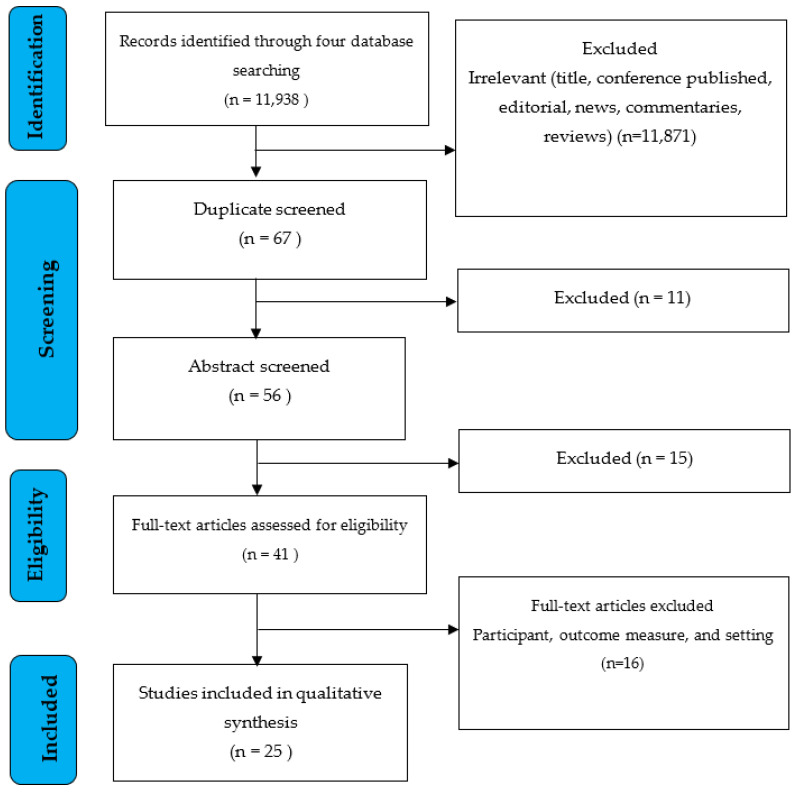
PRISMA flow diagram [35].

## Data Availability

All data generated or analyzed during this study are included in this published article [and its Appendix A].

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
