# Peer review of "The Use of Mobile Health Interventions for Outcomes among Middle-Aged and Elderly Patients with Prediabetes: A Systematic Review"

_ijerph, 2022, doi:10.3390/ijerph192013638_

Round 1

Reviewer 1 Report (Previous Reviewer 1)

Thank you for the opportunity to review the manuscript entitle The Use of Mobile Health Interventions for Outcomes Among Middle-Age and Elderly Patients with Prediabetes: A Systematic Review. This manuscript is very detailed and innovative. But I think there are still several areas that can be optimized. Below are my suggestions:

1. The authors say there are few systematic reviews about mobile health interventions from trial studies for middle-aged and elderly patients with prediabetes. However, it does not explain the differences and research gap between the previous reviews in this article, which should be explained in more detail highlighting how this review fills in some gap. Also, relevant research background needs to be supplemented in Introduction.

2. Although some tables and pictures in the article are detailed introductions to the literature or indicators, some of them are too tedious. It is recommended  simplifying them to improve the readability.

3. In the results section, such as "Dictionary Behaviors" and "Physical Activity", the author only focused on whether the results were statistically different without introducing the direction of change of the results, such as increased or decreased activity, which may cause confusion for readers. Therefore, the results of the research can be introduced in more detail.

4. I would suggest adding some discussion about strengths and limitations of the study.

Author Response

  1. Thank you for pointing this out. We agree with this comment. Therefore, we have change and can be found introduction section in third paragraph line 70-76.
  2. Thank you for this suggestion. We appreciate your input. It would have been interesting to explore this aspect. However, we disagree with your assessment that "some tables and pictures in the article are detailed introductions to the literature or indicators, some of them are too tedious". We believe it is clear and concise. We have tried to clarify where necessary. Would you please show in detail where you feel the structure is too tedious? We would appreciate your suggestions in order to improve the paper.

  3. Thank you for pointing this out. We agree with this comment. Therefore, we have changed, as can be found in the discussion sections

Reviewer 2 Report (Previous Reviewer 2)

As I suggested for the un-revised draft, this paper is a good suit for a conference proceeding, not as an academic article

Author Response

Thank you for the constructive comments on our manuscript. Would you please show in detail where you feel the draft is suitable for a conference proceeding? We would appreciate your suggestions in order to improve the paper.

Reviewer 3 Report (Previous Reviewer 3)

I am deeply appreciated for the effort of revision and I have no more major suggestion.

Only a minor suggestion for consideration: please consider to use other label rather than "V" to indicate inclusion in Table.

Author Response

Thank you for pointing this out. We agree with this comment. Therefore, we have change and can be found result section on table 4.

This manuscript is a resubmission of an earlier submission. The following is a list of the peer review reports and author responses from that submission.

Round 1

Reviewer 1 Report

This manuscript gathered and analyze information from experimental studies investigating the efficacy of mobile health usability for outcomes among middle age and elderly prediabetes patients. In below I included my comments:

1. The abbreviation of Type-2 Diabetes Mellitus may be "T2DM" instead of "TD2M" in the introduction section.

2.The discussion is very descriptive and any statements about the contribution and conclusions of the study are not deep. 

Reviewer 2 Report

The draft is suitable for a conference proceeding.

The overall structure in vague. The authors please try to modify and clarify where is necessary.

The reference is insufficient, at least some related popular articles that the paper need to cite.

The contents contain several huge tables, which is not appropriate for an academic journal.

Reviewer 3 Report

The manuscript proposes a systematic literature review of the intervention effect of mobile intervention on middle-aged and elderly patients with prediabetes. The focus of this paper is to understand the efficacy of mobile intervention on the prediabetes as an intervention form. I think the purpose of this research is very clear in the article, and it also has practical significance for the application and development of mobile intervention against diabetes. Although the purpose of the study is to explore the impact of mobile intervention forms on the intervention effect of patients, other variables that may cause interference are not presented in detail, so there is a certain impact on the reliability of the research conclusion.

In Part 3.4, the researchers divided the intervention forms into text messaging, mobile phone apps and Web-based interventions. However, even with the same intervention form, there are great differences in different studies. For example, Web-based interventions range from basic psychological education to risk scoring and individual counseling. The researchers did not further distinguish the differences in the details of the form of intervention, which may be one of the reasons why the research results cannot be consistent.

In Table 3, the scales or measurement methods used for different research measurement variables are not indicated, which reduces the reliability of the research conclusions.

Whether the control group of the intervention study was the active control group or the waiting group was not reported in the article.

The influence of the intervention content of different studies on the research results was ignored by the researchers.

The researchers did not clearly analyze the relationship between the intervention effect and the intervention form and other variables, which led to the conclusion content being vague and weak connection with the above. For example, there were contradictions in the intervention effects of many variables in different studies in the results, and the researchers lacked further explanation for these contradictions. In the conclusion part, the researchers proposed that the mobile intervention method has a positive effect on BMI, waist circumference and dietary habits, ignoring the contradictions in result part.

In addition, the benefits of using SMS as an intervention method and mobile health technology mentioned in the conclusion part seem to be more suitable for the introduction part.

In short, in the case of few relevant studies and inconsistent research results, the researchers did not conduct a more in-depth analysis of the variables that may affect the research results, but simply linked with the intervention forms of these researches, which reduced the value of this study.

In part 2.1, the search range of the literature is from January 2017 to July 2019, which is inconsistent with the time range mentioned in the abstract.